# The MSC-EV-microRNAome: A Perspective on Therapeutic Mechanisms of Action in Sepsis and ARDS

**DOI:** 10.3390/cells13020122

**Published:** 2024-01-09

**Authors:** Claudia C. dos Santos, Miquéias Lopes-Pacheco, Karen English, Sara Rolandsson Enes, Anna Krasnodembskaya, Patricia R. M. Rocco

**Affiliations:** 1Institute of Medical Sciences and Interdepartmental Division of Critical Care, Department of Medicine, University of Toronto, Toronto, ON M5B 1T8, Canada; 2Keenan Center for Biomedical Research, Unity Health Toronto, St. Michael’s Hospital, Toronto, ON M5B 1T8, Canada; 3Biosystems & Integrative Sciences Institute, Faculty of Sciences, University of Lisbon, 1749-016 Lisbon, Portugal; mlpacheco@fc.ul.pt; 4Laboratory of Pulmonary Investigation, Carlos Chagas Filho Institute of Biophysics, Federal University of Rio de Janeiro, Rio de Janeiro 21941-902, Brazil; prmrocco@gmail.com; 5Kathleen Lonsdale Institute for Human Health Research, Maynooth University, W23 F2H6 Maynooth, Ireland; karen.english@mu.ie; 6Department of Biology, Maynooth University, W23 F2H6 Maynooth, Ireland; 7Department of Experimental Medical Science, Faculty of Medicine, Lund University, 22184 Lund, Sweden; sara.rolandsson_enes@med.lu.se; 8Wellcome-Wolfson Institute for Experimental Medicine, Queen’s University of Belfast, Belfast BT9 7BL, UK; a.krasnodembskaya@qub.ac.uk; 9National Institute of Science and Technology for Regenerative Medicine, Rio de Janeiro 21941-599, Brazil; 10Rio de Janeiro Innovation Network in Nanosystems for Health-NanoSaúde, Research Support Foundation of the State of Rio de Janeiro, Rio de Janeiro 20020-000, Brazil

**Keywords:** acute lung injury, acute respiratory distress syndrome, extracellular vesicles, exosomes, mesenchymal stromal cells, microRNA, nano therapy, sepsis, systems biology

## Abstract

Mesenchymal stromal cells (MSCs) and MSC-derived extracellular vesicles (EVs) have emerged as innovative therapeutic agents for the treatment of sepsis and acute respiratory distress syndrome (ARDS). Although their potential remains undisputed in pre-clinical models, this has yet to be translated to the clinic. In this review, we focused on the role of microRNAs contained in MSC-derived EVs, the EV microRNAome, and their potential contribution to therapeutic mechanisms of action. The evidence that miRNA transfer in MSC-derived EVs has a role in the overall therapeutic effects is compelling. However, several questions remain regarding how to reconcile the stochiometric issue of the low copy numbers of the miRNAs present in the EV particles, how different miRNAs delivered simultaneously interact with their targets within recipient cells, and the best miRNA or combination of miRNAs to use as therapy, potency markers, and biomarkers of efficacy in the clinic. Here, we offer a molecular genetics and systems biology perspective on the function of EV microRNAs, their contribution to mechanisms of action, and their therapeutic potential.

## 1. Introduction

Sepsis and acute respiratory distress syndrome (ARDS) are complex and heterogeneous syndromes for which no specific therapies exist. Mesenchymal stromal cell (MSC) administration significantly reduces tissue inflammation and remodeling, improves pathogen clearance, and reduces morbidity and mortality in multiple preclinical models of sepsis and acute lung injury (ALI, the animal corollary to ARDS) [1]. Significant morbidity and mortality associated with both syndromes, as well as the lack of specific therapies, explains the urgency in demonstrating the therapeutic benefit of allogenic delivery of MSCs in the clinic. Although multiple studies have identified various targets that, when pharmacologically or interventionally altered, mitigate various pathological features of sepsis and ARDS experimentally (in vitro and in vivo) and in Phase I and II clinical trials, none has yielded effective treatments in Phase III randomized clinical trials.

Since the initial isolation and culture of human MSCs over 1300 registered clinical trials (clinicaltrials.gov “mesenchymal” 6/5/20) [2] have demonstrated the safety of administering these cells in humans [3]. Notwithstanding, we have yet to replicate the large effect sizes predicted from pre-clinical research [4,5], as small and large trials have failed to meet efficacy endpoints [6]. Inconsistent results have been attributed to product heterogeneity and irregularities, use of pre-clinical models that may not recapitulate the complexities of human disease, transferability across species, and/or poor estimation of effect size from preclinical data leading to inconclusive findings in humans [2,7]. Moreover, although the potential of MSCs remains undisputed, questions remain concerning their mechanisms of action (MoA). While understanding the MoA for a new drug is not required for its regulatory approval [8]—between 10% to 20% of currently Food and Drug Administration (FDA)-approved drugs have no known target or clear MoA [9]—the knowledge is a tool that can define the disorders they treat, unlock unknown disease processes, and enable biological and clinical advances to be made, well beyond phenomenological observations. Here, we specifically focus on the role of microRNAs (miRs) contained in MSC-derived extracellular vesicles (EVs). Evs are nanosized, membrane-bound vesicles released from cells that can transport a range of nucleic acids, lipids, and proteins necessary for cell-to-cell communication; herein lies their contribution to the MoAs and therapeutic potential of cell- and cell-free-based therapies.

## 2. MSCs Are Not Required for Therapeutic Effect in Sepsis/ALI

In preclinical studies of sepsis and ALI, MSCs delivered through a variety of different routes moderate multiple organ failure and reduce mortality [1]. From the initial studies, various investigators argued that although the precise MoA remains to be elucidated, cell engraftment with differentiation, transdifferentiation, or cell fusion were unlikely to contribute to observed beneficial effects in preclinical models of sepsis/ALI. First, very low levels of cell persistence could be demonstrated after infusion. Following intravenous administration, MSCs are efficiently targeted to the lung where they are trapped simply by size-based filtration in the distal pulmonary arteriolar bed [10]. Using highly sensitive methods (i.e., quantitative real-time polymerase chain reaction (RT-PCR) for the sex-determining region Y protein in female recipients [10,11,12]) or fluorescent labeling approaches [13,14,15], cells seem to be present in the lung only up to 28–96 h post-infusion. As reviewed by Gallipeau and others, after entrapment in the lung, nearly half of the infused MSCs are promptly phagocytosed by resident macrophages [16,17,18,19,20]. Masterson and colleagues have recently reviewed the literature on the distribution and fate of exogenously delivered MSCs, and while there are still important unknowns, it is thought that after disappearing in the lung, residual cells move to other major organs such as the liver and kidneys, where after a variable but short period (24 h to 14 days), they can no longer be detectable in the body [21].

Second, whole cells are not required for their therapeutic effect. Over a decade ago, it became apparent that the cultured supernatant from MSCs can mitigate lung injury by providing similar protection to that observed with viable whole cells [22,23], even when aerosolized [24]. In addition to confirming cell homing is not necessary for imparting therapeutic effects [25], this finding strongly supported a paracrine MoA. Multiple studies have also convincingly demonstrated that the immunomodulatory effect of MSCs is communicated via MSC and recipient-secreted cytokines and relies on the local microenvironment [19,26,27,28,29], as some of the observed effects depend on a pre-treatment of MSCs with inflammatory cytokines [27,28,30]. More recent findings indicate that the cytokine-mediated effects are only one part of the equation, as MSC-derived microparticles (including exosomes), apoptotic, metabolically inactivated, or even fragmented MSCs and MSC-membranes have immunomodulatory potential [31,32,33,34,35]. These observations have raised important questions regarding the putative direct targets of MSCs and the active therapeutic “ingredients” contained in an MSC, its secretome-derived “soup” or/and -derived microparticle/fragments, that confer their beneficial effects.

## 3. MSC-Derived Secretome

A large portion of MSC’s therapeutic activity is attributed to direct primary signaling through their secretome, comprising a multitude of cytokines, chemokines, growth factors, and subcellular vesicles; overviews of the current state of knowledge of MSC’s secreted mediators and how inflammatory priming influences their release have been published [36,37,38]. Target identification studies have demonstrated that MSCs have a wide pleiotropic effect on the pathophysiology of complex syndromes, specifically sepsis and ALI. Our group has previously published—using a model of polymicrobial sepsis induced by cecal ligation and puncture (CLP)—that MSC administration alters the expression of 3968 genes in five different sepsis-target organs (lung, liver, spleen, kidneys, and heart) [39]. Given about 30,000 as the estimated total number of genes in the mouse genome, our data suggest that approximately 13% of the septic genome is transcriptionally reprogrammed after MSC administration [39]. While multiple gain- (overexpressing) and loss- (silencing)-of-function studies have identified key mediators involved in the therapeutic effect, including IL-10 [40] and anti-bacterial peptide LL-37 [41], none have emerged as the single most responsible therapeutic target.

The release of EVs in the secretome represents an immediate cell communication mechanism that fulfills major criteria to impart MSC-related functions: (a) dissemination of membrane-bound mediators/signaling molecules; (b) transfer of expression patterns from one cell to another; and (c) rapid rearrangement of the cell surface. Like all messengers, EVs can not only immediately modify a phenotype of neighboring cells in a paracrine fashion but also bring forward an altered micro-milieu to other tissues by transferring the membrane composition and/or expression pattern to distal organs [42]. Salutary as well as pathological effects may be imparted by information contained on EVs and their cargo in the form of circulating nucleic acids (mRNAs and miRs), lipids, and proteins [31,43,44].

Seminal work from Monsel and colleagues demonstrated the therapeutic effects of human MSC-derived EVs in a mouse model of severe pneumonia [45]. In this model, the group was able to show that EV delivery significantly enhanced the expression of keratinocyte growth factor (KGF) in recipients associated with improved survival. A subsequent clinical trial randomized patients with ARDS to receive recombinant KGF or placebo. The trial was stopped early because of increased harm in the group that received KGF [46]. While multiple reasons are usually linked to unsuccessful clinical trials, results did suggest that the increase in KGF is a meta-phenomenon rather than directly causal to the improved mortality seen in animal models. Herein lies the difficulty and the critical importance of understanding and translating promising preclinical data to the clinic. Together, these data challenge the paradigm that a single, specific MSC-derived paracrine mediator is responsible for the global pleiotropic effect of MSCs on transcriptional and network reprogramming in sepsis. A much more plausible explanation is that MSC-conferred protection from sepsis includes a range of complementary activities, resulting in the mitigation of the innate and acquired immune and inflammatory responses.

Of particular interest to our group is the potential therapeutic role of regulatory RNAs contained within EVs. Below, we reviewed the literature implicating epigenetic regulatory small RNAs, specifically miRs, as important regulators and mediators of the therapeutic effect of MSC-derived EVs.

## 4. MSC-EV Content

MSC-derived EVs act as paracrine mediators of MSCs’ beneficial effects. In a landmark publication [47], Phinney and colleagues demonstrated that MSCs are inefficient in performing mitophagy; using microvesicles to shuttle mitochondrial components out of the cell. As MSCs outsource mitophagy to recipient macrophages [47], they simultaneously package anti-inflammatory miRs into smaller EVs (including microRNA let-7f, c and I, 23b, 27b, and 29a). These miRs were shown by the group to inhibit critical pattern and damage recognition receptors and signaling molecules: Toll-like receptor (TLR)4, MyD88, TLR9, TLR7, and tumor necrosis factor-alpha (TNF-α) in recipient macrophages. There is little doubt EVs contain anti-inflammatory miRs, that are produced and released by MSCs; what is unclear is why the cells would do this, and whether these miRs are actually therapeutically active?

The fact that RNAs contained inside EVs may impart biological function was first described in a seminal publication by Valadi and colleagues [48]. This group demonstrated the presence of mRNA from approximately 1300 genes in exosomes of endocytic origin, many of which were not present in the cytoplasm of donor cells. Taking advantage of cross-species-specific sequences in functional genomic experiments, these authors demonstrated RNA from mast cell exosomes were transferable from mouse to human mast cells and that after transfer, new mouse proteins were found in the recipient cells, indicating that transferred exosomal mRNAs occurs and are functional. The presence of Argonaute 2 (AGO2, an essential component of the RNA-induced silencing complex [RISC]) and miR inside purified Evs remains controversial [49], but studies indicate that following endocytosis, EV-associated miRs are loaded onto the host cell Argonaute proteins, where the release of internalized miRs can alter cellular function [50]. In addition, while associated with the miRISC/AGO2 complex, internalized miRs can bind to their complementary sequences inside a cell (driving cell–cell communication), modulating cellular responses. Accordingly, many companies are now exploring miR-based therapeutics. Miravirsen (produced by Roche/Santaris) and RG-101 (produced by Regulus Therapeutics), designed to treat hepatitis C, are considered the flagship products of this class of future drugs [51,52,53,54,55,56]. A successful trial of a miR-221 selective inhibitor for the treatment of refractory advanced cancer has recently demonstrated the excellent safety profile, promising bio-modulator, and anti-tumor activity, offering a rationale for further clinical investigation [57].

Much of the controversy around which component of EV payload is biologically active (contributes to MoA) comes from non-standardized EV preparations. The importance of this issue cannot be understated [58,59,60]. The International Society for EVs has established clear criteria for isolating, characterizing, and defining microparticles derived from MSCs [61,62]. Moreover, Dr. Sai Kiang Lim and colleagues have recently published criteria for defining MSC-derived EVs for therapeutic application, including establishing MSC origin (concentration of CD73, CD90, and CD105; and absence of non-MSC antigens CD14, CD34, and CD11b); number of particles per unit weight protein/membrane lipids and particle diameter (within 50–200 nm); molar/weight ratio of protein to membrane lipids (e.g., cholesterol and phosphatidylcholine); and presence of biochemically active cargo (enzyme activity of CD73, unit activity per μg protein) [63]. The fact that MSC-derived EVs and miRs are therapeutically relevant is not disputed; the issue is whether the miRNAs that are carried inside EVs are actually the ones imparting their beneficial effects.

## 5. EV-microRNAome

In the publication from Valadi and others, the authors determined that each exosome may carry as many as 100–120 different miRs and that some miRs were expressed to a greater extent in exosomes compared to donor cells [48]. Baglio and colleagues further demonstrated that MSCs of different origins have similar small RNA expression profiles; EV-derived miRs (EV-miRs) or exo-miRs represent about 1–2% of the cellular RNA content, and multiple highly expressed miRs are precluded from EV sorting [64], suggesting that the process of packaging miRs into EVs is highly regulated and not simply a mechanism for managing cellular waste. Quantitative and stoichiometric analysis of the miR content of EVs demonstrated less than one copy of each microRNA per EV [65]. Using ultrasensitive total internal reflection-based single-vesicle in situ quantitative and stoichiometric analysis of tumor-derived EV miRs, He and colleagues demonstrated that each EV may contain as many as 6 copies of different unique microRNAs [66].

But what is the functional dose of a miR? Based on the literature, the functional dose may differ for each target [67]; with factors such as the expression levels of the targets, the number of miR-binding sites on the target mRNA and feedback loops able to shift this narrow window for effective gene suppression. MiRs may also repress their targets in a nonlinear manner introducing thresholds in gene expression [68]. Furthermore, as miRs can work cooperatively or competitively [69], dose-dependent mRNA target selection becomes even more complicated when combinations of miRs, with potentially overlapping targets, are taken into account, such as would happen in EVs carrying a variety of different miRs. Electroporation nano straw delivery, which can bypass biological mechanisms, has been used to deliver miRs with precise dosage control directly into primary cells [70]. An example of the importance of dose comes from studies using miR-17-92, which was shown to decrease the cell viability of colon cancer cell line at low doses (0.00003 μg plasmid) but increase cell viability at high doses (0.3 μg plasmid) [67]. MiR fold changes as low as 3–4 are sufficient to drive the development of disease in transgenic mice [71], and a 1.5-fold change in the expression level of a single miR is considered enough to alter phenotype.

Denzler and colleagues showed that when a miR is lowly expressed, only the highest-affinity sites are sufficiently occupied to mediate repression, but as miR expression increases, more and more intermediate and low-affinity sites have occupancies sufficient to mediate repression [72]. Canonical 6-nucleotide (-nt) sites, which typically mediate modest repression, can nonetheless compete for miR binding, with a potency of 20% of that observed for canonical 8-nt sites. This suggests there is a strong relationship between site affinity and miR-dose that is significantly impacted by competition between different miRs for overlapping binding sites. Moreover, cooperative binding of proximal sites for the same or different miRs does increase potency, making it very difficult to understand the relationship between dose and biological effect.

In most clinical studies, EV doses range between 1–10 × 10^8–12^ particles/mL, suggesting that while the copy number inside an EV may be small, the EV dose may deliver a significant number of therapeutically relevant miRs. Moreover, given that a single miR can regulate the expression of hundreds of genes, a system biologist’s view of miR networks suggests that the biological effects of a single functioning miR may significantly amplify the biological effect at a systems level (Figure 1) [73].

## 6. Evidence That miRs Transferred in MSC-Derived EV Are Functional

Accumulating evidence suggests that miR transfer in MSC-derived EVs plays an important role in attenuating sepsis-associated lung injury by exerting anti-inflammatory and anti-apoptotic effects in both in vitro and in vivo models of lung disease. Specifically, in models of septic and non-septic ALI, several miRs (miR-21-5p, miR-27a-3p, miR-30b-3p, miR-100, miR-384-5p, miR-145, miR-146, miR-125b-5p, and miR-181a-5p, among others) transferred in MSC-derived EVs have been identified to be involved in therapeutic actions (Table 1).

Macrophages are key innate immune cells involved in ARDS inflammatory response, and many studies have investigated the ability of EVs to modulate macrophage inflammatory state. MiR-145 has been shown to account for the human bone marrow MSC-derived EV effects in the model of *E.coli*-induced lung injury. It has been demonstrated that miR-145 packaged in EVs is transferred into murine macrophages to enhance macrophage phagocytosis and to reduce *E. coli* bacterial load through increasing leukotriene B4 levels [74]. Moreover, miR-145 is also responsible for modulating ABCC1, an ATP-binding cassette multidrug resistance protein [74]. Song and colleagues found that miR-146a was strongly upregulated in MSCs by IL-1β licensing and selectively packaged into EVs. This group showed that miR-146a was transferred to macrophages, resulting in M2 polarization leading to increased survival in septic mice [75]. Inhibition of miR-146a in MSCs partially negated the immunomodulatory properties of MSC-derived EVs [75]. Similarly, Yao and colleagues demonstrated that macrophages are polarized into M2 phenotype by miR-21 transferred from MSC-derived EVs pre-treated with IL-1β, which significantly enhanced the therapeutic effects in experimental sepsis [76]. These macrophages demonstrated decreased production of IL-8 and TNF-α, increased expression of IL-10, and enhanced phagocytosis activity [77,78].

In the mouse model of ischemia/reperfusion lung injury, the anti-apoptotic miR-21-5p in MSC-derived EVs was transported to alveolar macrophages, contributing to macrophage reprogramming toward M2 macrophages via phosphatase and tensin homolog (PTEN) and programmed cell death protein 4 (PDCD4) [79]. Such effects were reversed when MSCs were pretreated with miR-21-5p antagonists, resulting in persistent apoptosis of pulmonary endothelial cells [79]. Wang and colleagues have shown that miR-27a-3p packaged in MSC-derived EVs was taken up by alveolar macrophages and directly targeted nuclear factor kappa B (NF-κB) subunit 1, thereby downregulating NF-kappa B signaling and promoting M2 macrophage polarization in the murine LPS-induced model of lung injury [80]. Lentiviral transduction of MSCs with anti-miR-27a-3p or knockdown of miR-27a-3p in vivo abolished the therapeutic effects of MSC-derived EVs in that model [80].

Ferroptosis is a unique modality of cell death driven by iron-dependent phospholipid peroxidation. Shen and colleagues reported that adipose tissue MSC-derived EV transfer of miR-125b-5p rescues pulmonary microvascular endothelial cells ferroptosis and improves survival in a CLP model of sepsis. Such effects were mediated via the Keap1/Nrf2/GPX4 pathway [81]. Furthermore, Pei and colleagues recently reported that human-umbilical-cord-MSC-derived EVs prevent inflammation and reduce the severity of lung injury in the mouse model of sulfur mustard-induced lung injury through the delivery of miR-146a-5p [82]. In particular, they showed that miR-146a-5p expression levels were markedly decreased after injury, while it was restored in lung tissue after treatment with MSC-derived EVs. Moreover, through a series of elegant experiments, the authors deciphered that miR-146 targeted TRAF6 and mediated the anti-inflammatory effect through modulation of TLR4, TRAF6, IRAK1, and NF-κB pathways. Overexpression of miR-146a-5p in MSC-derived EVs significantly attenuated TRAF6 expression, which negatively regulated sulfur mustard-induced inflammation in vitro and in vivo [82]. MiR-451 is another factor that is highly abundant in MSC-EVs [83]. This miR was found to inhibit the expression of IL-1β, IL-6, and TNF-α by suppressing the TLR4/NF-κB signaling pathway in burn-induced ARDS. Such effects were abrogated when miR-451 expression was suppressed [83]. Furthermore, both miR-23a-3p and miR-182-5p carried by MSC-EVs were able to reverse LPS-induced injury and fibrosis by silencing Ikbkb and destabilizing IKKβ, which prevented the downstream activation of NF-κB and hedgehog pathways [84].

MSC-derived EVs overexpressing miR-30b-3p were protective in the mouse model of LPS-induced lung injury presumably through inhibition of serum amyloid A3 (SAA3). This factor is a major component of the acute phase of inflammation and is a direct target of miR30b-3p. In MLE-12 cells (mouse alveolar type II epithelial cell line), miR30b-3p transferred from MSC-derived EVs protected cells against LPS-induced injury and enhanced cellular proliferation [85]. Furthermore, miR-132-3p carried by MSC-derived EVs reduced LPS-induced injury in mouse lung epithelial cells by targeting TRAF6, inhibiting PI3K/Akt signaling and enhancing cellular proliferation [86]. In endothelial cells, MSC-EV-derived miR-126 induced downregulation of the Sprouty-related EVH1 domain-containing protein 1 (Spread-1), which prevented LPS-induced injury and enhanced cell function [87]. MiR-126 was also involved in inhibiting the expression of the alarmin HMGB1, which increased the expression of tight junction proteins [88]. miR-150 transferred from MSC-derived EVs was also able to mitigate LPS-induced endothelial injury by regulating caspase-3, Bax-Bcl-2, and MAPK signaling [89]. When miR-150 antagomirs were transfected into MSCs, these immunomodulatory effects were partly reversed [89].

Although the exact role of autophagy in lung injury remains controversial, it may be a double-edged sword depending on the underlying causes of lung injury and the stages of disease progression. In this context, Chen and colleagues have demonstrated that miR-100 is at least partially responsible for the therapeutic effects of Wharton’s Jelly MSC-derived EVs in the rat model of bleomycin-induced lung injury via enhancement of autophagy through inhibition of mTOR [90]. MiR-100 overexpression in MSC-EVs reduced alveolar epithelial cell apoptosis, total protein content, neutrophil counts, and levels of the proinflammatory cytokines in the bronchoalveolar lavage fluid (BALF) of bleomycin-injured rats [90]. In contrast, Liu et al. found that bone marrow MSC-derived EVs could ameliorate LPS-induced ALI in rats by inhibiting autophagy stress in alveolar macrophages via miR-384-5p [91]. This miRNA was able to impair autophagy by directly targeting beclin-1, thus alleviating LPS-induced injury [91]. On the other hand, MSC-derived EVs can transfer miR-223/142 to inhibit the activation of the NLRP3 inflammasome and prevent the activity of M1 pro-inflammatory macrophages [92].

In our recent study, we analyzed the miR profile of MSC-derived EVs after MSCs were exposed to BALF samples from ARDS patients [93]. MiR-181a-5p was found to be among the most highly enriched in the MSC-derived EVs. Further investigation of its functional role showed that miR-181a-5p transfer resulted in human and murine alveolar macrophage reprogramming towards an anti-inflammatory state via the PTEN-STAT5-SOCS1 axis. More importantly, we demonstrated that this pathway was activated in human monocyte-derived macrophages in the presence of plasma from ARDS patients [93]. We also demonstrated that in the model of LPS-induced lung injury, EVs lacking miR-181a were not able to reduce lung injury and inflammatory cell infiltration into the lungs compared to control EVs. Notably, administration of control EVs was coupled with significant upregulation of pSTAT5 and SOCS1 expression in alveolar macrophages in vivo. Furthermore, miR-181a overexpression resulted in significant augmentation of the therapeutic efficacy of EVs in this model [93]. While gain and loss of function reconstitution experiments provide evidence that EV-miRs may be more than just an epiphenomenon, the causal link to MoA is re-enforced by evidence that the parent-MSC macro- and micro-milieu modifies EV-miR content and function.

**Table 1 cells-13-00122-t001:** Biological activity of miRNAs carried by MSC-derived EVs in the in vitro and in vivo models.

MicroRNA	Main Biological Activity	Reference
miR-145	↑ macrophage phagocytosis activity	[74]
miR-146a	↑ macrophage polarization to M2 anti-inflammatory phenotype	[75]
miR-21	↑ macrophage polarization to M2 anti-inflammatory phenotype	[76]
miR-21-5p	↑ macrophage polarization to M2 anti-inflammatory phenotype	[79]
miR-27a-3p	↓ NF-kappa B signaling, ↑ macrophage polarization to M2 anti-inflammatory phenotype	[80]
miR-125b-5p	↓ ferroptosis-induced inflammation by regulating Keap1/Nrf2/GPX4 expression	[81]
miR-146a-5p	↑ anti-inflammatory actions by targeting TLR4, TRAF6, IRAK1, and NF-κB pathways	[82]
miR-451	↓ IL-1β, IL-6, and TNF-α levels by suppressing the TLR4/NF-κB pathway	[83]
miR-23a-3p/miR-182-5p	↓ injury and fibrosis by silencing Ikbkb and destabilizing IKKβ	[84]
miR-30b-3p	SAA3 inhibition, ↑ proliferation, and ↓ apoptosis of alveolar epithelial cells	[85]
miR-132-3p	PI3K/Akt signaling inhibition, ↑ cellular proliferation	[86]
miR-126	↓ Spread-1 expression, ↑ endothelial cell activity, ↓ HMGB1 expression, ↑ tight junction protein expression	[87,88]
miR-150	↓ endothelial injury by regulating caspase-3, Bax-Bcl-2, and MAPK signaling	[89]
miR-100	↓ alveolar epithelial cell apoptosis	[90]
miR-384-5p	↓ autophagy stress in alveolar macrophages	[91]
miR-223/142	↓ NLRP3 inflammasome activation	[92]
miR-181a-5p	↑ macrophage polarization to M2 anti-inflammatory phenotype via the PTEN-STAT5-SOCS1 axis	[93]

## 7. Licensing of MSCs Alters EV miR Content and Therapeutic Activity

The requirement for licensing or activation for MSCs to mediate their immunomodulatory or cytoprotective effects is now better understood [28]. In particular, simulation with IFN-γ, TNF-α, IL-1β or combinations of these cytokines enhances MSC therapeutic efficacy in vitro and in vivo [14,94,95]. Other pro-inflammatory agents, including MIF [96], TGF-β1 [97], LPS [98], Poly:I.C. [99], and Pam3CSK4 [100], as well as pharmacological agents (e.g., eicosapentaenoic acid [101,102]) or biologically relevant samples (e.g., serum and BALF [20,34,43]), have been shown to enhance MSC immunosuppressive and protective functions. Exposure to hypoxia has also been used to prime MSCs leading to improved MSC survival and increased production of anti-inflammatory and cytoprotective mediators [103,104]. The therapeutic activity of MSC-derived EVs can also be enhanced using various alternative licensing approaches [28].

Pro-inflammatory cytokine or thrombin stimulation of MSCs leads to increased EV production [105,106,107]. Several licensing strategies reliably enhance MSC-derived EV immunoregulatory and therapeutic effects. These priming strategies include pre-exposure to TNFα, IL-1β, IFN-γ alone or in combination [28,75,76,105,108,109,110,111,112,113,114,115,116,117]; TLR ligands such as LPS [118,119,120], hypoxia [121,122], or other factors, including thrombin [107], the endoplasmic reticulum stress inducer, thapsigargin [123], and the lipophilic photosensitizer, hexyl-5-aminolevulinate hydrochloride (HAL) in combination with pro-inflammatory cytokines [113]; transfection with miR [124]; or serum from injury models [20,34,43], BALF, as above; or secretome from LPS-activated cells such as microglia [125]. EVs derived from licensed or primed MSCs have been investigated in an array of inflammatory conditions and while the pathogenic factors driving specific diseases may differ, there is a large consensus across studies demonstrating the protective and immunoregulatory effects of MSC-derived EVs. Much attention has been centered on the relevance of priming strategies on macrophage polarization [43,75,76,105,109,113,116,119,120,121,122,123,125] or transition toward pro-resolution of infection or inflammatory cellular phenotypes [112,118].

Mechanistically, studies have elegantly identified active components contained within MSC-derived EVs and targeted signaling cascades involved. IL-1β priming of MSCs significantly enhanced MSC-derived EV protection in CLP models of sepsis [75,76] by suppressing macrophage activation and promoting M2 polarization, leading to reduced expression of TNF-α, iNOS and increased expression of IL-10 and ARG-1 [75,76]. These elegant studies identified different miRs in primed MSC-derived EVs. Elevated levels of miR-146a were identified in IL-1β-primed MSC-derived EVs and were shown to contribute to the immunomodulatory effects of IL-1β-primed MSC-derived EVs using miR-146a mimics and inhibitors [75]. A role for miRNA has also been identified in IFN-γ-primed MSC-derived EVs in the context of dextran sodium sulfate-induced colitis. EVs secreted by IFN-γ-primed MSCs showed superior efficacy compared to control MSC-derived EVs. Inhibiting miR-125a and miR-125b decreased IFN-γ-primed MSC-derived EVs’ capacity to suppress Th17 cell differentiation. The importance of miR-125a and miR-125b was verified using agomirs to abrogate beneficial effects [114].

While the promotion of M2 macrophages is a common effect mediated by primed MSC-derived EVs in inflammatory conditions, it also features in regenerative models. In comparison to normoxic MSC-derived EVs, hypoxic (1% O_2_ culture) MSC-derived EVs exerted greater effects in promoting liver regeneration associated with M2 polarization [122]. MiR-182-5p was significantly enriched in hypoxic MSC-exosomes, and inhibition of miR-182-5p partially abolished the protective effects. In this setting, M2 polarization involved Foxo1/TLR4 signaling [122]. These studies demonstrate that whatever therapeutic “ingredients” are being packaged into an EV by MSCs, this process is dose-responsive, suggesting it may be further manipulated and tailored for therapeutic purposes.

## 8. Generation of EVs with Favorable Characteristics by Genetically Modifying the Parental MSCs: Role of EV-miR Content

EVs from lineage-defined MSCs induce lineage-specific changes in target naïve MSCs [126,127]. The miR cargo of EVs is representative of its parental cell status, and the differentiated state of the parental MSC confers lineage-defining properties to the derivative EVs [128,129]. Shirazi and colleagues demonstrated the importance of cellular and exosomal miRs to MSC functions by knocking down both Dicer and AGO2 in parental MSCs. Dicer/AGO2 deficient cells markedly reduced the differentiation potential of MSCs. Treatment of Dicer/AGO2-deficient MSCs with wild-type MSC-derived exosomes effectively recovered the impaired osteoblastic differentiation. Dicer/AGO2 knockdown reduced the quantity and diversity of miRs present in MSC-exosomes. MiR sequencing and KEGG analysis implicated the miR-dependent effects on multiple osteoinductive pathways in Dicer/AGO2 deficient cells. Importantly, elegant miR replacement experiments, where transfection of Dicer/AGO2-deficient MSCs with mimics of miRs significantly downregulated in Dicer/AGO2-deficient knockdown cells, were shown to recover/reconstitute functions of exosome-mediated signaling in MSCs [130], providing robust and compelling in vitro evidence of the critical contribution of exosome-derived miRs to the biological function of EVs. Whether exomiRs, miRs contained within exosomes or EVs, are also functional in vivo can be inferred from pre-clinical studies that have exploited genetic engineering strategies to modify the activity and therapeutic potential of EVs.

MSCs have been genetically engineered to overexpress several genes [28,131,132]. These include the first report using angiopoietin 1 [133], heme oxygenase (HO)-1 [134], superoxide dismutase (SOD) [135], IL-10 [136,137,138], IL-1RL1 [139], angiotensin-converting enzyme (ACE)2 [140,141], vascular endothelial growth factor (VEGF) [142], KGF [143], and hepatocyte growth factor (HGF) [137,144], among others. Compared to unmodified MSCs, genetically engineered MSCs promoted a greater reduction in inflammation and tissue injury and improved survival when administered in animal models of ALI or sepsis. This approach has also been used to engineer and enrich EVs with specific properties. Both gain and loss of function may be exploited. In a recent model of myocardial dysfunction, rat adipose-derived MSC-overexpressing stem cell factor (SCF) was successfully used to induce endogenous regenerative processes and improve cardiac function. EVs from genetically modified MSCs were isolated, and a total of 95 differentially expressed miRNAs were identified in intact GFP- or SCF-overexpressing rat MSC EVs [145]. Genetic modification of MSCs can greatly impact EV miR composition [145].

Gómez-Ferrer and colleagues demonstrated that EVs derived from a genetically modified MSC line overexpressing hypoxia-inducible factor 1-alpha (HIF-1α) and telomerase suppressed the proliferation of activated T-cells more effectively than unmodified MSCs [146]. In another study by Gong and colleagues, it was demonstrated that EVs produced by MSCs overexpressing GATA-4 promoted angiogenesis by increasing the tube-like structure formation and spheroid-based sprouting of human umbilical vein endothelial cells (HUVECs) compared to EVs obtained from control MSCs (MSCs transduced with an empty vector) [147]. Vieira and colleagues modified MSCs to overexpress miR-135b or miR-210 using a lentiviral vector and found that the modified MSCs secreted EVs with different miR signatures. EVs enriched with miR-210 demonstrated an improved angiogenesis potential, as assessed by tube formation assays, as well as increased angiogenesis-related proteins in HUVECs (including e.g., Activin A and Endothelin-1) compared to control EVs [148]. Genetic engineering has also been used to enhance the production of EVs with improved stability or enhanced target abilities. For example, by expressing a cardiomyocyte-binding peptide (Lamp2b fused to a cardiomyocyte-specific peptide) on the EV surface, Mentkowski and colleagues demonstrated increased delivery of cardioprotective cargo to cardiomyocytes [149].

Various strategies have also been used to modify the parental cells or EV themselves to enhance their therapeutic potential, including (i) modifying the EV surface by fusing a peptide that targets a specific receptor on the recipient cell with an EV-enriched protein (e.g., Lamp2b); (ii) via chemical conjugation such as metabolically labeled protein or targeting peptide; or (iii) enhancing the EV cargo release using, for example, a pH-sensitive fusion peptide expressed on the EV membrane [150]. Certain proteins in the recipient cell may also play a major role in EV uptake. For example, the expression of heparan sulfate proteoglycans has been reported to be essential for the uptake of cancer-cell-derived exosomes [151]. Similarly, inhibition or knockout of sialic acid-binding immunoglobulin-type lectin resulted in decreased exosome uptake by the recipient cell [152]. There is increasing evidence that EV glycosylation plays an important role in their uptake by the recipient cells [153]; however, there is still a great knowledge gap with respect to the EV glycosylation processes and how genetic modifications of the parental MSCs could be used to obtain EVs with favorable characteristics. Further investigation is required to clarify these issues.

Regarding the role of EV-miRs and the role of genetic engineering as a strategy to tailor EV-miR content for therapeutic purposes, compelling evidence comes from Huag and colleagues [129]. In a non-sepsis-related model, this group evaluated if pathway-specific EV engineering can be accomplished via targeted expression of select miRs in MSC EVs. To accomplish this, the group generated MSCs that constitutively expressed miR-424 in an EV-specific manner. This was achieved by using sequence motifs present in miRs that control their localization into exosomes. The protein heterogeneous nuclear ribonucleoprotein A2B1 (hnRNPA2B1) specifically binds exosomal miRs through the recognition of specific motifs and controls miR loading into exosomes [154]. The loading of miRs into exosomes can therefore be modulated by mutagenesis of identified motifs or changes in hnRNPA2B1 expression levels. Preferential targeting of the miR of choice—in this case, miR-424—was achieved via infection with a lentivirus containing a dual promoter vector encoding for the miR-424 followed by the EV targeting sequence, resulting in overexpression of miR-424 linked to the exosome localization sequence in parental genetically modified MSCs. Results showed that miRs can be specifically expressed in EVs without affecting the physical EV characteristics as well as their ability to be endocytosed by target cells. The EVs engineered displayed enhanced differentiation potential of naïve MSCs compared to controls, and over-representation of miR-424 was able to trigger osteogenic differentiation by specifically enhancing the SMAD1/5/8 pathway that governs BMP2 signaling in vivo. These studies demonstrate proof-of-concept of the enormous potential for this genetic engineering approach to exploit different miRs for therapy.

## 9. Concluding Remarks

In this review, we offer a perspective on how important questions regarding the contribution of EV-miRs to appropriate function may be reconciled with the low copy number, pleotropic effects, and competing and cooperative activities, using evidence from molecular genetics and systems biology. Genetic experiments presented offer compelling evidence for the role of EV-miRs in mediating therapeutically relevant MoA. A systems biologist’s view of the presence of cooperative and competing miR activity provides an explanatory vision of how even low abundance miRs may significantly impact miR networks and function in recipient cells. The ability to tailor EV-miR content either by licensing or genetically modifying parent cells opens exciting new avenues for advancing personalized care using cell-free technologies. How we will translate this exciting new technology to clinical practice remains to be elucidated. The first in-human trial of genetically modified MSCs for the treatment of sepsis is currently underway (NCT04961658); results will be critical in defining how this technology will evolve for the application of EV technologies for the treatment of sepsis and ARDS.

## Figures and Tables

**Figure 1 cells-13-00122-f001:**
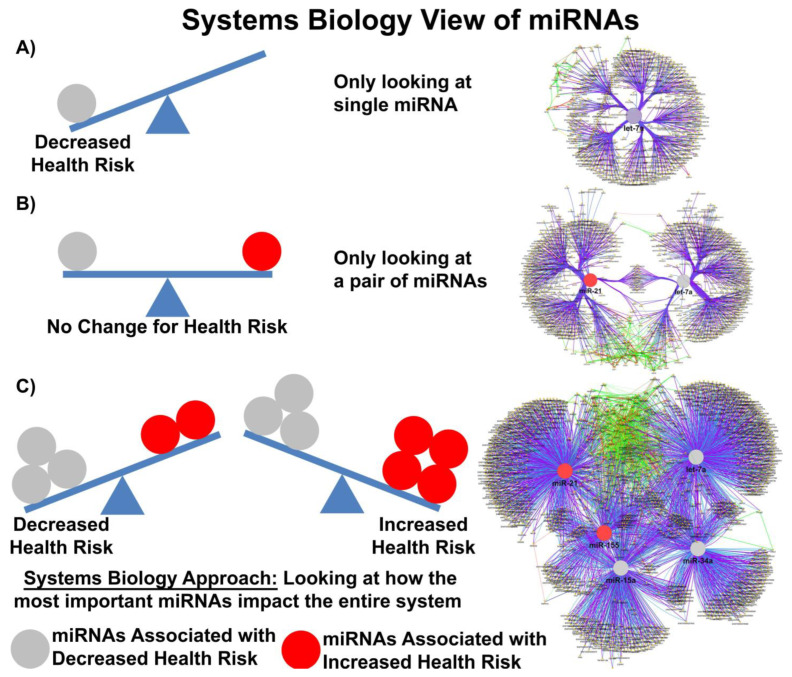
Systems biology view of microRNA Networks. Figure generated with Cytoscape and ClueGO by Dr. Afshin Beheshti, Bioinformatician and Principal Investigator, Blue Marble Space Institute of Science, NASA Ames Research Center, Space Biosciences Research Branch, reproduced with permission from Dr. Beheshti and NASA Ames Research Center.

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
