# Peer review of "The MSC-EV-microRNAome: A Perspective on Therapeutic Mechanisms of Action in Sepsis and ARDS"

_cells, 2024, doi:10.3390/cells13020122_

Round 1
Reviewer 1 Report
Comments and Suggestions for Authors
Claudia C. dos Santos and colleagues present an excellent revision focused on the therapeutic mechanisms of action of the microRNA of MSCs-derived EVs for sepsis and ARDS. The take-home-message and information included are clear and of relevance in its field. The manuscript is suitable for publication.
I only migh recommend the consideration to insert a small sentence to define EVs when it first appears in the manuscript.
Author Response
Response to reviewers:
Reviewer 1: a small sentence to define EVs when it first appears in the manuscript.
R: Thank you. We have added one sentence when EVs first appear:
"Here, we specifically focus on the role of microRNAs (miRs) contained in MSC-derived extracellular vesicles (EVs). EVs are nanosized, membrane-bound vesicles released from cells that can transport a range of nucleic acids, lipids and proteins necessary for cell-to-cell communication; herein lies their contribution the MoAs and therapeutic potential of cell- and cell-free-based therapies."
Reviewer 2 Report
Comments and Suggestions for Authors
1. Sepsis can damage the lungs, kidneys, liver and other organs, this review paper gives the reviewer the impression it is more focused on lung injury (Sepsis induced and ARDS). If there are publications and trials that demonstrate efficacy of MSC-EV in kidney, liver and/or heart, the authors may want to include more samples.
2. Texts above Fig.1 mentioned dose in clinical studies, it needs in depth discussion on off-target in case of high dose. The authors also should give some examples of health risk, e.g. some observations of adverse effects reported in trials.
3. Fig.1 appears to be very unbalanced, schematic demonstration to the left is a bit oversized, as well the texts in the figure; the right side shows a lot of information, the readers might want to have some level of description to understand these busy plots. In case of (c), I think it deserves some discussion on how the switching between "decreased" & "increased" risk is extracted from the figure to the right.
Author Response
We thank the reviewer for his/her insightful comments. Please find a point-by-point response.
1. Sepsis can damage the lungs, kidneys, liver and other organs, this review paper gives the reviewer the impression it is more focused on lung injury (Sepsis induced and ARDS). If there are publications and trials that demonstrate efficacy of MSC-EV in kidney, liver and/or heart, the authors may want to include more samples.
R: Yes, we agree with the reviewer – the group of authors includes primarily lung experts and the review is very focused on lung. However, there are 2 main reasons why we think focusing on other organs is not relevant for this review: First, the focus of the current review is to understand the contribution of microRNAs to the therapeutic action of EVs – rather than reviewing the literature on pre-clinical or clinical EV studies. The studies we chose to highlight, were selected because they reveal something about how scientists have gone about developing experiments to understand the mechanistic contributions of EV-miRs, to provide evidence that the EV-miR content is not random and is therapeutically relevant. Making the review more comprehensive would be good but would detract from the central purpose and message of this perspective. The second reason, a bit more pragmatic, is that the review is already very long. The topic, microRNA in sepsis induced organ failure, is very broad and will require more than one piece of work to catalogue and unravel the biology. On that note, we have recently submitted a second comprehensive review looking at the role of therapeutic microRNAs in ARDS to a special issue of IJMS Contribution to Special Issue “The Role of microRNA in Human Diseases 2.0” – entitled “Acute Respiratory Distress Syndrome: microRNAs and microRNA-based Therapeutics” and a third comprehensive review paper that was submitted to Oxygen MDPI entitled “Pre-clinical Studies of MicroRNA-based Therapies for Sepsis: A Scoping Review” where we focused on the role of microRNAs in regulating oxidative stress during sepsis.
- Texts above Fig.1 mentioned dose in clinical studies, it needs in depth discussion on off-target in case of high dose. The authors also should give some examples of health risk, e.g. some observations of adverse effects reported in trials.
R: Again, we completely agree with the reviewer. This is a fundamental question that has very important clinical implications to the field and the future of EV-based therapeutics – but again – the topic is so broad and so important that we feel this warrants a dedicated review to carefully review the pre-clinical and clinical data and extract the critical data regarding dose and off-target effects. Here we focused on the basic principles of dose and how it pertains to mechanism of action:
But what is the functional dose of a miR? Based on the literature, the functional dose may differ for each target [67]; with factors such as the expression levels of the targets, the number of miR-binding sites on the target mRNA, and feedback loops being able to shift this narrow window for effective gene suppression. MiRs may also repress their targets in a nonlinear manner introducing thresholds in gene expression [68]. Furthermore, as miRs can work cooperatively or competitively [69], dose-dependent mRNA target selection becomes even more complicated when combinations of miRs, with potentially overlapping targets, are taken into account – such as would happen in EVs carrying a variety of different miRs. Electroporation-nano straw delivery, which can bypass biological mechanisms, has been used to deliver miRs with precise dosage control directly into primary cells [70]. An example of the importance of dose comes from studies using miR-17-92, which was shown to decrease cell viability of colon cancer cell line at low doses (0.00003 μg plasmid) but increase cell viability at high doses (0.3 μg plasmid) [67]. MiR fold changes as low as 3-4 are sufficient to drive the development of disease in transgenic mice [71] and a 1.5-fold change in the ex-pression level of a single miR is considered enough to alter phenotype.
Denzler and colleagues showed that when a miR is lowly expressed, only the highest-affinity sites are sufficiently occupied to mediate repression, but as miR ex-pression increases, more and more intermediate - and low-affinity sites have occupancies sufficient to mediate repression [72]. Canonical 6-nucleotide (-nt) sites, which typically mediate modest repression, can nonetheless compete for miR binding, with a potency of 20% of that observed for canonical 8-nt sites. This suggests there is a strong relationship between site affinity and miR-dose that is significantly impacted by com-petition between different miRs for overlapping binding sites. Moreover, cooperative binding of proximal sites for the same or different miRs does increase potency – making it very difficult to understand the relationship between dose and biological effect.
In most clinical studies, EV doses – range between 1-10 x 108-12 particles/mL, suggesting that while the copy number inside an EV may be small, the EV dose may de-liver a significant number of therapeutically relevant miRs. Moreover, given that a single miR can regulate the expression of hundreds of genes, a system biologist’s view of miR networks suggests that the biological effects of a single functioning miR may significantly amplify the biological effect at a systems level (Figure 1) [73].
- Fig.1 appears to be very unbalanced, schematic demonstration to the left is a bit oversized, as well the texts in the figure; the right side shows a lot of information, the readers might want to have some level of description to understand these busy plots. In case of (c), I think it deserves some discussion on how the switching between "decreased" & "increased" risk is extracted from the figure to the right.
R: The figure was reproduced from Dr. Afshin Beheshti’s seminal discussion for the Blue Marble Space Institute of Science, NASA Ames Research Center – the point of the figure is not the specifics of the disease or the specifics of each microRNA – the figure would obviously be completely different for specific case (of disease or miRs). The point is to provide a schematic interpretation – from a system’s biology perspective - of how differential miR regulation results in the competing regulation of competing networks – resulting in systems regulation – this is a vision of miR-based system’s biology that helps explain how even a single miR copy can have a profound effect on the system, thus providing an explanation for how to reconcile the stochiometric issue of low copy numbers of miRNAs with their biological effects.